# Robust Detection of Adversarial Attacks by Modeling the Intrinsic Properties of Deep Neural Networks

**Zhihao Zheng**
Department of Computer Science
Brandeis University
Waltham, MA 02453
zhihaozh@brandeis.edu

**Pengyu Hong**
Department of Computer Science
Brandeis University
Waltham, MA 02453
hongpeng@brandeis.edu

## Abstract

It has been shown that deep neural network (DNN) based classifiers are vulnerable to human-imperceptive adversarial perturbations which can cause DNN classifiers to output wrong predictions with high confidence. We propose an unsupervised learning approach to detect adversarial inputs without any knowledge of attackers. Our approach tries to capture the intrinsic properties of a DNN classifier and uses them to detect adversarial inputs. The intrinsic properties used in this study are the output distributions of the hidden neurons in a DNN classifier presented with natural images. Our approach can be easily applied to any DNN classifiers or combined with other defense strategies to improve robustness. Experimental results show that our approach demonstrates state-of-the-art robustness in defending black-box and gray-box attacks.

## 1 Introduction

Since the successful application of deep convolutional neural network to large-scale image recognition [26] by Krizhevsky et al. [11], neural network based Deep Learning has gained significant attentions. Researchers have been shown that deep neural networks (DNN) were able to deliver state-of-the-art performances in various fields, such as, robotics [20, 7], self-driving cars [1, 2], face recognition for identification [33], games [29, 30, 19], biomedical image processing [6, 28], and so on. Despite of these successes, DNN-based classifiers have a severe weakness [34]. For example, knowing the architecture and parameters of a DNN classifier (i.e., white-box attack), an adversarial example can be easily constructed to fool the DNN classifier by applying a small perturbation to an input image. Even though the perturbations are too small to affect human recognition, the DNN classifier can misclassify the perturbed input with high confidence. Successful attacks can also be black-box, in which the architecture and parameters of a DNN are unknown to the attackers [23]. Interestingly, adversarial images remain malicious even after printed out and then fed to a well trained DNN [12]. A variety of algorithms have been developed to generate powerful attacks [34, 9, 12, 24, 32, 4, 21, 22].

Without proper safeguards, users of DNN-based applications can be exposed to unforeseen hazardous situations caused by "trivial" noises. Various attempts have been conducted to defend adversarial attacks. Papernot et al. [25] proposed a defensive distillation approach, which reduces the magnitude of gradients during training, to make the trained model more robust to input perturbations . However, it was later shown that the distillation approach was still highly vulnerable to attacks [3]. Recently, the adversarial training strategy became popular [9, 18, 13]. This strategy augments the training data with adversarial examples to enhance the capability of DNNs to deal with targeted attacks. It focuses more on defending black-box attacks, and usually does not consider white-box attacks. Hence, it is still vulnerable to iterative attacks [13]. In addition, the robustness of the model trained by this strategy depends on the attacks covered by the adversarial training examples. Another strategy is

to actively detect adversarial inputs [17, 15] by training a two-class classifier that takes the hidden states of a DNN as the input to tell if an input is adversarial. Unfortunately, this strategy can seriously suffer from attacks unseen during the training procedure.

In this paper, we propose a strategy for detecting adversarial inputs by modeling the intrinsic properties of a DNN classifier. This strategy does not need to know the attack methods or require to train the classifier with adversarial samples. Therefore, it will suffer less from unseen attacks. We implement an approach, termed I-defender ("I" stands for intrinsic), which explores one of the intrinsic properties of a DNN classifier, i.e., the distributions of its hidden states given natural training data. We reason that: When the DNN classifier mis-assigns a specific class label to an adversarial input, its hidden states are quite different from those given natural data of the same class. I-defender models the hidden state distributions of a DNN classifier given natural data and uses them to detect adversarial inputs. We do not attempt to model the hidden state distributions of a classifier presented with adversarial inputs because such distributions will depend specifically on the attack methods that generate the adversarial training inputs. For the same reason, we do not try to build a model to distinguish the hidden state distributions of a classifier presented with natural inputs from those of the classifier presented with adversarial inputs. We do not try to model the input distribution because the input space is of much higher dimension. In addition, the hidden state distributions encode the generalization power of the model.

## 2 Related Works

### 2.1 Adversarial Attacks

**Fast Gradient Sign Method (FGSM)** [9] efficiently generates adversarial examples. The perturbation $\rho$ is computed along the direction in the input space which maximally increases a linearized cost function under $\ell_\infty$ norm:

$$\rho = \epsilon \cdot sign(\nabla J(\theta, \mathbf{x}, l)) \tag{1}$$

where $\epsilon$ is a scalar to restrict the norm of the perturbations, $\nabla J$ is the gradient of the cost function, and $\mathbf{x}$ is the original input with its true class label as $l$. The perturbation generated by this method usually is small with respect to the maximum value of the input.

**Basic Iterative Method (BIM)** [12] is an iterative method. At each iteration, it adds a small perturbation decided by the gradient $\nabla J$ at the current version of the perturbed input and clips the modifications in the range of $\epsilon$ from the original input.

$$\mathbf{x}_{adv}^{i+1} = clip_\epsilon\{\mathbf{x}_{adv}^i + \alpha \cdot sign(\nabla J(\theta, \mathbf{x}_{adv}^i, l))\} \tag{2}$$

where $\mathbf{x}_{adv}^i$ denotes the perturbed input generated at the $i$-th iteration. Usually, $\alpha$ is set to 1, and the number of iterations is set to 10. Based on this approach, Metzen et al. [17] developed another iterative method, which produces each perturbation in the direction of $\ell_2$-normalized gradient and project the perturbed version back to the $\epsilon$ ball around the original input if the $\ell_2$ distance between them exceeds $\epsilon$.

$$\mathbf{x}_{adv}^{i+1} = project_\epsilon\{\mathbf{x}_{adv}^i + \alpha \frac{\nabla J(\theta, \mathbf{x}_{adv}^i, l)}{||\nabla J(\theta, \mathbf{x}_{adv}^i, l)||_2}\} \tag{3}$$

**DeepFool Method** [21] iteratively computes the minimal norm adversarial perturbation. Inputs are assumed to reside in a region confined by the decision boundary of a classifier. In the $i$-th iteration, the classifier is linearized around the perturbed input $\mathbf{x}_{adv}^i$. The algorithm will find the closest class boundary and take the minimal step to traverse the boundary accordingly to the $l_p$-norm distance from $\mathbf{x}_{adv}^i$. The perturbations are accumulated to the original input until mis-classification is achieved. DeepFool is able to achieve the same successful level of attack as FGSM while using smaller perturbations.

### 2.2 Perturbation Detection Methods

Metzen et al. [17] proposed to augment a DNN with a subnetwork that focuses on adversarial perturbation detection. The subnetwork connects each layer of the DNN and is trained separately

using a dataset containing both natural samples and adversarial samples generated by known attack methods. Although this method can achieve certain degree of success, the trained subnetwork can be easily fooled by adversarial examples generated by attack methods unseen during training [15]. Lu et al. [15] introduced SafetyNet that trains a Support Vector Machine [5] to detect the boundary between natural and perturbed data in the space of the quantified features from a DNN. Similar to the above subnetwork approach, SafetyNet is trained using certain attacking methods. Although it may produce more robust results, it still suffers from unseen adversarial patterns. Samangouei et al. [27] proposed Defense-GAN to leverage the expressive capability of Generative Adversarial Network [8] to defend against attacks. Defense-GAN trains a generative model to model the distribution of natural inputs. To detect adversarial inputs, Defense-GAN projects an input onto the range of the GAN generator by a Gradient Descent (GD) procedure to minimize the Wasserstein distance between the input and the sample generated by the GAN generator. The generator runs the GD procedure several time with different seed inputs. An input will be detected as an attack if the minimal Wasserstein distance between the input and the generated samples is larger than a threshold. To achieve a higher accuracy, Defense-GAN needs to try more seed inputs and run more GD iterations, which will be time-costly. Its performance also relies on the quality of its GAN, which can be challenging to train for complex tasks.

## 3 I-defender

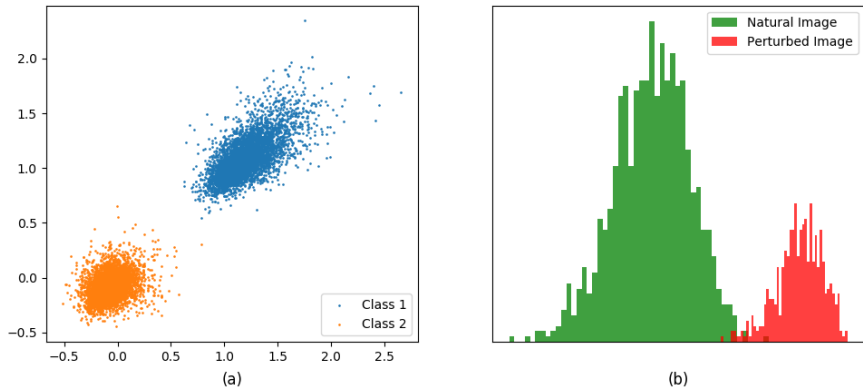

Figure 1: Hidden state distribution examples. The architecture of the DNN is specified in Section 4.2. The DNN was trained on CIFAR-10. (**a**) The IHSDs of two classes (class 1: automobile and class 2: bird) are plotted in the 2D subspace with the largest variances. (**b**) The intrinsic distribution (green) of a hidden state of the airplane class *versus* the distribution (red) of the same hidden state when the DNN misclassifies perturbed inputs as airplane.

Szegedy et al. [34] interpreted adversarial examples as the low probability pockets in the manifold which are never or rarely seen during training. There are too many of them so that attackers can easily explore them to fool the trained classifier. Typically, adversarial attacks can be generated by traversing the high dimension decision space along the direction of gradient to reach those pockets. Adversarial training can be viewed as a method to fill those pockets with adversarial examples generated by known attack methods. However, when dealing with complex applications, this strategy may not be effective because there can be infinite number of low probability pockets. Instead of trying to fill those pockets, Defense-GAN tries to model the input distribution, which however can be too complex to deal with. On the other hand, given natural data, the distributions of the hidden states (i.e., outputs of hidden neurons) of a DNN classifier can be much simpler (e.g., Figure 1a). The dimensions of the hidden state spaces are often much lower than that of the input space, which can make the hidden state distributions much easier to model than the input distribution. We call a hidden state distribution of a DNN presented with natural data as an intrinsic hidden state distribution (IHSD), which characterizes certain intrinsic properties of the DNN. I-defender uses the IHSDs of a classifier to reject adversarial inputs because they tend to produce hidden states lying in the low density regions of the IHSDs (e.g., see Figure 1b). I-defender can be easily attached to any models that produce internal representations.

In our current implementation, I-defender uses Gaussian Mixture Model (GMM) to approximate the IHSD of each class as the following:

$$p(\mathcal{H}(x)|\theta, c) = \sum_{k=1}^{K} w_i \mathcal{N}(\mathcal{H}(x)|\mu_{ck}, \Sigma_{ck}) \tag{4}$$

where $\mathcal{H}(x)$ denotes the hidden state of an input $x \in$ the $c$-th class, $\theta$ denotes the DNN classifier of interest (or its parameters), and $\mu_{ck}$ and $\Sigma_{ck}$ are the mean and covariance matrix of the $k$-th Gaussian component in the mixture model of the $c$-th class. After training the DNN classifier, we feed all training samples into it and collect the corresponding hidden states for training a GMM for each class using the EM algorithm [16].

In this study, all DNN classifiers consisted of convolutional layers followed by fully connected layers. The states of the convolutional layers are position-dependent, which make it non-trivial to model directly. Thus, we choose to only model the state of the fully connected hidden layers. For each class $c$, a threshold $TH_c$ is chosen to reject inputs by checking if their likelihoods are lower than $TH_c$.

$$Reject(x, c) = p(\mathcal{H}(x)|\theta, c) < TH_c \tag{5}$$

## 4 Experiments

We evaluated I-defender on standard datasets (MNIST, F-MNIST, CIFAR-10) against several attack methods including $\ell_\infty-$ norm Iterative, $\ell_2-$ norm Iterative, FGSM, and DeepFool. The number of iterations was set to 10 for iterative attack methods ($\ell_\infty$ and $\ell_2$). The results are organized accordingly to the following attack types:

1. Black-box Attack: Attackers know nothing about the defense strategy and use a substitute network to generate adversarial samples.

2. Semi White-box Attack: Attackers know all details of the DNN classifier, but however have no knowledge of its defense strategy. Attackers use the gradients from the DNN to generate adversarial samples.

3. Gray-box Attack: Attackers know the architecture of the DNN classifier and its defense strategy, but have no knowledge of their parameters. Attackers use a network of the same architecture and the same defense strategy to generate adversarial samples.

### 4.1 Black-box Attack

In this experiment, we used two datasets: the MNIST dataset [14] and the F-MNIST dataset [35], a more challenging replacement of the MNIST dataset. It was shown that the Defense-GAN detection method outperformed previous approaches on these two datasets under the FGSM attack. Hence, we first compared I-defender only with the Defense-GAN detection under the FGSM attack. We used the same experiment settings to those used by in the Defense-GAN detection experiment [27]. Attackers generated adversarial examples using model E, and used them to attack model F (see Table 1 for details of models E and F). The results are summarized in Tables 2-5 (the results of the Defense-GAN detection are copied from [27]).

Table 1: Architectures of Models E and F. The architectures are the same to those used in the adversary detection experiment in [27]. FC($n$) denotes a fully connected layer with $n$ neurons. Conv($k, w \times h, s$) denotes a convolutional layer with $k$ output features, filter size of $w \times h$ and stride as $s$. ReLU is the Rectified Linear Unit activation.

| Model E | Model F |
|---|---|
| FC(200) | Conv( $64, 8 \times 8, 2$ ) |
| ReLU | ReLU |
| FC(200) | Conv( $128, 6 \times 6, 2$ ) |
| ReLU | ReLU |
| FC(10)+Softmax | FC(10)+Softmax |

Table 2: I-defender $vs$ Defense-GAN of different settings. The FGSM attack used $\epsilon = 0.3$. The MNIST data was used.

| Method | Detection AUC | Number of GD runs | Iteration number in each GD run |
|---|---|---|---|
| **I-defender** | 0.993 | N/A | N/A |
| Defense-GAN | 1.0 | 10 | 800 |
| Defense-GAN | 1.0 | 10 | 400 |
| Defense-GAN | 0.985 | 10 | 50 |
| Defense-GAN | 0.982 | 5 | 100 |
| Defense-GAN | 0.922 | 2 | 100 |
| Defense-GAN | 0.836 | 1 | 100 |

Table 3: I-defender $vs$ Defense-GAN (10 GD runs and 400 iterations in each run) under the FGSM attack on the MNIST data. The detection AUC is used as the measurement. The ROC curves of I-defender are shown in Figure 2

| $\epsilon$ | Defense-GAN | I-defender |
|---|---|---|
| 0.1 | 0.914 | 0.964 |
| 0.15 | 0.975 | 0.979 |
| 0.2 | 0.989 | 0.988 |
| 0.25 | 0.998 | 0.991 |
| 0.3 | 0.999 | 0.993 |

Table 4: I-defender $vs$ Defense-GAN of different settings under the FGSM attack with $\epsilon = 0.3$. The F-MNIST data was used.

| Method | Detection AUC | Number of GD runs | Iteration number in each GD run |
|---|---|---|---|
| **I-defender** | 0.985 | N/A | N/A |
| Defense-GAN | 0.987 | 10 | 800 |
| Defense-GAN | 0.983 | 10 | 400 |
| Defense-GAN | 0.965 | 10 | 100 |
| Defense-GAN | 0.945 | 5 | 100 |
| Defense-GAN | 0.935 | 10 | 25 |
| Defense-GAN | 0.876 | 2 | 100 |
| Defense-GAN | 0.794 | 1 | 100 |

Table 5: I-defender $vs$ Defense-GAN (10 GD runs and 200 iterations in each run) under the FGSM attack. The F-MNIST data was used. The detection AUC is used as the measurement. The ROC curves of I-defender are shown in Figure 3

| $\epsilon$ | Defense-GAN | I-defender |
|---|---|---|
| 0.1 | 0.775 | 0.9302 |
| 0.15 | 0.884 | 0.9587 |
| 0.2 | 0.940 | 0.9722 |
| 0.25 | 0.969 | 0.9807 |
| 0.3 | 0.985 | 0.9850 |

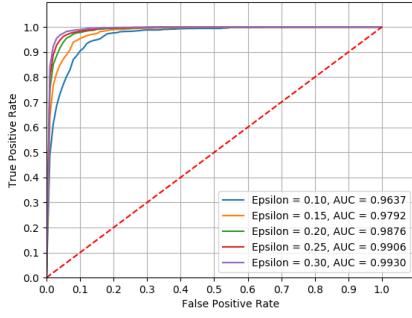

Figure 2: The ROC curves of I-defender attacked by FSGM with different $\epsilon$ on the MNIST dataset.

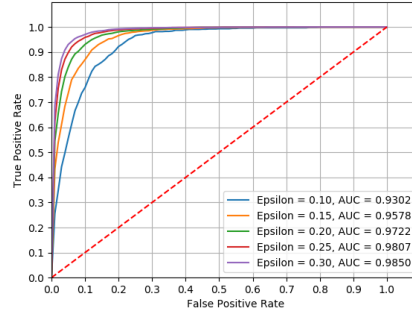

Figure 3: The ROC curves of I-defender attacked by FSGM with different $\epsilon$ on the F-MNIST dataset.

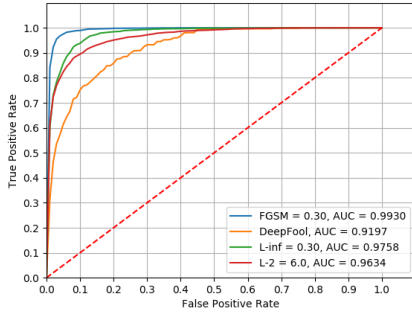

Figure 4: The ROC curves of I-defender attacked by various methods on the MNIST dataset

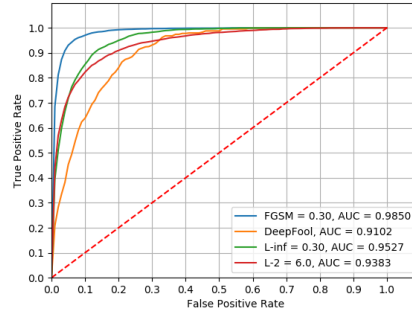

Figure 5: The ROC curves of I-defender attacked by various methods on the F-MNIST dataset.

The performance of Defense-GAN detection highly depends on its hyper-parameters (i.e., the number of GD runs and the number of GD iterations). Tables 2 and 4 show that I-defender outperforms Defense-GAN with less than 10 GD runs and less than 400 GD iterations. Although Defense-GAN is able to produce a slightly better result than I-defender by trying more GD runs (e.g., 10) and more GD iterations (e.g., 800), we estimated that the running time of Defense-GAN (reported in [27]) was $10^5$ times slower than I-defender. Tables 3 and 5 summarize the effects of changing the attacking power ($\epsilon$) of FGSM, and the corresponding ROC curves are shown in Figures 2 and 3. I-defender suffers much less than Defense-GAN when the perturbation level is more subtle. We think this is due to that I-defender models the hidden state distributions while Defense-GAN models the input distribution. It is explained in [9] that small input perturbations can be amplified across layers and cause a hidden state to grow by $\omega^T \eta$, where $\omega$ means weight vector and $\eta$ means perturbation. Thus, even if a perturbation is small, the hidden states can be altered significantly and be easily detected by I-defender. We also tested I-defender using other attacking methods. The ROC curves and AUC values show that I-defender is robust (see Figures 4 and 5).

## 4.2 Semi White-box Attack

In this experiment, we used the CIFAR-10 dataset [10], which is more complex than MNIST and F-MNIST. We trained a 34-layer wide residual network [36] with k = 8 as the classifier. Since Defense-GAN did not report their results on CIFAR-10, we compared I-defender with two supervised detection methods, SafetyNet[15] and Subnetwork[17]. The experiment settings (e.g., the attack methods, the attack strengths, the balanced data for evaluate detection accuracy, etc.) were the same to those reported in [15]. The results are summarized in Table 6. Supervised methods perform better against the attack methods/strengths that are used in their training phases. Their performances drop

significantly when facing unseen attack methods/strengths. I-defender performs more consistently across different attack methods/strengths, and significantly outperforms supervised methods on unseen attack methods/strengths.

Table 6: Semi white-box attack results measured by detection AUC. The results of SafetyNet (SVM and mSVM) and Subnetwork are incorporated from [15]. The column headers denote the attack methods (Iter-$\ell_\infty$: Iterative-$\ell_\infty$; Iter-$\ell_2$: Iterative-$\ell_2$) and their attack strengths indicated by "Adv Acc" (i.e., accuracy of classifier on classifying adversarial samples). The parameters of the attacking methods were set to match with the Adv Acc's specified in [15]. Both SafetyNet and Subnetwork were trained by Iterative $\ell_\infty$ with Adv Acc = 13.14% (i.e., the attack in the first column), and then were generalized to other attacks in the rest columns. The results of I-defender ID-95 and ID-99 were obtained by setting the likelihood thresholds to keep 95% and 99% of the natural training data, respectively.

| Method | Iter-$\ell_\infty$ | Iter-$\ell_2$ | FGSM | DeepFool | Iter-$\ell_\infty$ | FGSM |
| Adv Acc | 13.14% | 10.80% | 27.84% | 24.53% | 29.56% | 45.68% |
|---|---|---|---|---|---|---|
| SVM | 83.6 | 84.840 | 75.545 | 78.305 | 76.330 | 61.755 |
| mSVM | 92.52 | 93.915 | 74.480 | 86.635 | 76.670 | 60.895 |
| Subnet | 98.235 | 98.660 | 68.980 | 49.270 | 49.295 | 49.27 |
| ID-95 | 81.28 | 79.398 | 87.464 | 87.636 | 78.730 | 85.024 |
| ID-99 | 79.04 | 74.173 | 90.485 | 90.709 | 74.580 | 87.596 |

## 4.3 Gray-box Attack

In a gray-box attack, an attacker knows the structure and defense strategy of a DNN classifier, but has no knowledge of its parameters. Adversarial training is able to increase the robustness of a DNN classifier when attacked by single-step gray-box methods, but not by iterative gray-box methods. To test the robustness of I-defender under gray-box attacks, we trained two deep networks with same structure independently on the same natural training data, and tested I-defender on three attack methods: FGSM, Iterative $\ell_\infty$, and Iterative $\ell_2$. We let the attackers maximize the following function to derive a perturbed input $x$,

$$\arg\max_x J(x, y_s|\theta) + \alpha \prod_{k \neq s} Reject(x, y_k) \times \max_{k \neq s} \log P(\mathcal{H}(x)|y_k) \qquad (6)$$

where the first term $J(.)$ denotes the cross entropy loss of classifying $x$ into its true label $y_s$ by a classifier with parameter $\theta$ (this term encourages finding a perturbation $x$ that will lead to misclassification), and the second term penalizes $x$ if it is detected by our defense mechanism (i.e., it is not accepted by any classes other than $y_s$ because its likelihoods from any other classes are lower than the corresponding thresholds). The second term will encourage moving $x$ towards the most promising class other than its true class $y_s$.

Our first experiment was on CIFAR-10, and the results (Tables 7 and 8) indicate the following. Although an attacker can successfully exploit the weakness in an I-defender, which is fully exposed to the attacker, the attacker can hardly affect the detection power of another I-defender whose parameters are unknown to the attacker. This is because two classifiers are trained independently and can have very different hidden state spaces/distributions. Moreover, the change of DNN architecture (more complex ones) should not greatly influence the performance of our approach. We also carried out an experiment on ImageNet by setting $\alpha$ to 1. Without deploying I-defender, the adversarial accuracy of a "Target" DNN can easily drop to around 10%. I-defender made it significantly harder for an attacker to succeed (Table 9). In addition, the lower the adversarial accuracy of the "Target" DNN, the higher the detection AUC. This indicates that I-defender can be used to effectively defend attacks. We think this is because a DNN architecture appropriate for a more difficult task (e.g., ImageNet) is sophisticated and has many distinct local minimals. Hence, if trained twice independently, it will produce two DNN instances with very different hidden state spaces/distributions. Therefore, a perturbed input generated accordingly to a "Source" DNN can easily lead to a remarkably different hidden state configuration in a "Target" DNN, which means it is challenging for attackers to rely on a "Source" DNN to successfully attack a "Target" DNN.

Table 7: Defense against gray-box attacks on CIFAR-10. Adversarial examples were generated from the "Source" DNN (WRN34-8 [36]) and tested on both the "Source" and "Target" DNNs (WRN34-8). Performances are measured by detection AUC.

| $\alpha$ | Iterative-$\ell_\infty$ ($\epsilon = 0.013$) Target / Source | Iterative-$\ell_2$ ($\epsilon = 0.47$) Target / Source |
|---|---|---|
| 1e-3 | 85.203 / 76.450 | 86.249 / 75.786 |
| 1e-2 | 83.261 / 58.967 | 84.740 / 69.243 |
| 1e-1 | 83.875 / 59.459 | 85.406 / 71.179 |
| 1 | 83.945 / 59.554 | 85.576 / 72.027 |
| $\alpha$ | Iterative-$\ell_\infty$ ($\epsilon = 0.0085$) Target / Source | FGSM ($\epsilon = 0.075$) Target / Source |
| 1e-3 | 86.130 / 74.463 | 94.472 / 93.564 |
| 1e-2 | 85.485 / 63.151 | 94.480 / 93.387 |
| 1e-1 | 85.777 / 63.598 | 94.450 / 93.359 |
| 1 | 85.745 / 63.590 | 94.457 / 93.355 |

Table 8: Evaluate the effects of DNN size on the performance of I-defender. We compared WRN34-8 and WRN46-8 in this experiment.

| $\alpha$ | Iterative-$\ell_\infty$ ($\epsilon = 0.013$) WRN-34 / WRN-46 | Iterative-$\ell_2$ ($\epsilon = 0.47$) WRN-34 / WRN-46 |
|---|---|---|
| 1e-3 | 85.203 / 87.549 | 86.249 / 86.23 |
| 1e-2 | 83.261 / 87.555 | 84.740 / 86.215 |
| 1e-1 | 83.875 / 87.637 | 85.406 / 85.844 |
| 1 | 83.945 / 87.553 | 85.576 / 85.528 |
| $\alpha$ | Iterative-$\ell_\infty$ ($\epsilon = 0.0085$) WRN-34 / WRN-46 | FGSM ($\epsilon = 0.075$) WRN-34 / WRN-46 |
| 1e-3 | 86.130 / 85.960 | 94.472 / 94.060 |
| 1e-2 | 85.485 / 86.084 | 94.480 / 94.159 |
| 1e-1 | 85.777 / 85.969 | 94.450 / 94.174 |
| 1 | 85.745 / 86.113 | 94.457 / 94.111 |

Table 9: Defense against gray-box attacks on ImageNet. Both the "Source" and "Target" DNNs used VGG19 [31]. Note that the accuracy of the "Target" DNN on the natural data is 71.028%. The "Adv Acc" represents the adversarial accuracy of the "Target" DNN under attacks. Performances are measured by detection AUC.

| $\alpha$ | FGSM ($\epsilon = 0.3$) Adv Acc / AUC | Iter-$\ell_\infty$ ($\epsilon = 0.019$) Adv Acc / AUC | Iterative-$\ell_2$ ($\epsilon = 12, 5$) Adv Acc / AUC |
|---|---|---|---|
| 1e-2 | 70.892 / 62.685 | 62.636 / 97.201 | 59.032 / 93.832 |
| 1e-1 | 70.892 / 62.801 | 62.646 / 97.211 | 59.016 / 93.837 |
| 1 | 70.892 / 62.612 | 62.648 / 97.215 | 59.010 / 93.834 |

# 5 Conclusion and Discussion

We show that modeling the intrinsic properties of a DNN classifier can be a reliable strategy to detect adversarial attacks. This strategy does not need any knowledge about attack methods. Hence, it does not suffer from attack methods unseen during training and is able to robustly defense against various of black-box and gray-box attacks, which is sufficient for most application scenarios. Our implementation of this strategy uses GMM to approximate the hidden state distribution of a DNN classifier. Experiment results validate that our implementation achieves state-of-the-art performance among unsupervised methods and generalizes better than supervised ones. Our method is straightforward and can be easily incorporated into any DNN-based classifiers and can also be easily combined with any existing defense strategies. Depending on applications, one can replace GMM with other more appropriate models to approximate hidden state distributions. Since our method models the hidden states of a DNN instead of the inputs, it can be directly applied to other modalities (such as text).

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
