[Reviews · NeurIPS 2018]

Reviewer 1



The authors propose a novel method for detecting adversarial attacks on neural network classifiers by modelling the distribution of outputs of hidden fully connected layers rather than the distribution of inputs which is commonly done. The method is called I-defender and it is compared with other methods on an extensive set of benchmarks. The distribution of outputs of the hidden layers is modelled as a mixture of Gaussians which is fit using the EM algorithm. I'm not at all familiar with the literature on adversarial attacks so I can only write a review assuming the overview of related work in the submission is accurate. The paper is very clearly written and does a good job of explaining its contributions. The experiments are sensible and show clearly show excellent performance of the suggested approach, often beating the state-of-the-art methods. The idea of modelling the distribution of hidden layers instead of inputs makes a lot of sense to me and if it is indeed novel then it can have a lot of impact. One general concern I have regarding the approach of detecting adversarial examples by modelling the distribution of real data is that innocent inputs from a different distribution than the training set (such as digits written in different style) could potentially be misclassified as adversarial attacks. This issue is not mentioned at all in the paper. However, since I'm not familiar with the literature I don't know if it's fair to expect the reviewers to address it.

Reviewer 2



This work focuses on an approach to detect adverserial inputs to a given DNN. The authors' approach involves understanding the output distribution of the neurons in a DNN in response to the natural intended input. They argue that this distribution will be different from the distribution produced by adverserial input. The authors model the distribution of the hidden states through Gaussian Mixture Model (GMM). The decision whether an input is natural/adverserial is based on a threshold placed on the distribution. The merit of the approach is explored empirically for Black-Box, Semi White-Box and Grey-Box adverserial scenarios. The authors show that at times their approach can outperform baselines. They provide some reasoning (such as speed, performace on unseen samples etc,) for still using their approach where other baselines might be better. It would be interesting to know how the performance changes with change in DNN architecture. Do wider/deeper networks have any influence on the performance. For very large and complex models, is it possible to use only layers close to the input/output and still be able to achieve decent performance. How about bigger tasks such as ImageNet classification. How about other domains apart from images? Currently, GMM is used to model the neuron distribution. Does it make a difference if the model of hidden neuron distribution is changed?

Reviewer 3



The paper presents an unsupervised learning approach to the problem of adversarial attack detection in the context of deep neural networks. The authors model the intrinsic properties of the networks to detect adversarial inputs. To do so, they employ a Gaussian Mixture Model (GMM) to approximate the hidden state distribution, in practice the state of the fully connected hidden layers, and detect adversarial samples by simply checking that their likelihood is lower than a given threshold. Exhaustive experimental results in different show that the proposed method achieves state-of-the-art performance compared to unsupervised methods while generalizing better than supervised approaches. The paper reads well and is technically sound. The method is very simple but seems to work well. The experiments are rather convincing. I don’t have any major concern but I think that it would be interesting to see how the methods work with more complex networks (e.g. with more than 10 classes) and with other models than GMMs. I have a couple of comments/questions: - For the sake of repeatability, it would be good to have more details about the EM procedure (number of iterations?) - In section 3/table 6, it is indicated that the likelihood threshold is set to keep 95 and 99% of training data. How is the threshold selected for the other experiments (section 4.1)? minor comments: - Figure 2: “The OC curves”- “The ROC curves” - Figure 3 is not mentioned in the text - Line 154: “even the perturbations”-> “even if the perturbations” - Line 166” “outperform”->”outperforms” - Line 173: “same on the training”->”on the same training”